# EvoFed: Leveraging Evolutionary Strategies for Communication-Efficient Federated Learning

**Mohammad Mahdi Rahimi**    **Hasnain Irshad Bhatti**    **Younghyun Park**
**Humaira Kousar**    **Jaekyun Moon**
KAIST
{mahi,hasnain,dnffkf369,humairakousar32}@kaist.ac.kr
jmoon@kaist.edu

## Abstract

Federated Learning (FL) is a decentralized machine learning paradigm that enables collaborative model training across dispersed nodes without having to force individual nodes to share data. However, its broad adoption is hindered by the high communication costs of transmitting a large number of model parameters. This paper presents EvoFed, a novel approach that integrates Evolutionary Strategies (ES) with FL to address these challenges. EvoFed employs a concept of 'fitness-based information sharing', deviating significantly from the conventional model-based FL. Rather than exchanging the actual updated model parameters, each node transmits a distance-based similarity measure between the locally updated model and each member of the noise-perturbed model population. Each node, as well as the server, generates an identical population set of perturbed models in a completely synchronized fashion using the same random seeds. With properly chosen noise variance and population size, perturbed models can be combined to closely reflect the actual model updated using the local dataset, allowing the transmitted similarity measures (or fitness values) to carry nearly the complete information about the model parameters. As the population size is typically much smaller than the number of model parameters, the savings in communication load is large. The server aggregates these fitness values and is able to update the global model. This global fitness vector is then disseminated back to the nodes, each of which applies the same update to be synchronized to the global model. Our analysis shows that EvoFed converges, and our experimental results validate that at the cost of increased local processing loads, EvoFed achieves performance comparable to FedAvg while reducing overall communication requirements drastically in various practical settings.

## 1   Introduction

Federated Learning (FL) provides a decentralized machine learning framework that enables model training across many devices, known as clients, without needing to collect and process sensitive client data on the centralized server [1]. The typical FL process begins with each client downloading an identical initialized model from a central server, performing model updates with local data, and then uploading the updated local model for the next communication round. Subsequently, the server combines the uploaded models to refine the global model, typically using a technique like FedAvg [2]. This iterative cycle repeats for a fixed number of rounds, ensuring collaborative model improvement across clients.

Although FL provides notable benefits, such as a certain level of privacy preservation and the utilization of diverse data sources, one of the major challenges associated with FL is the significant

communication overhead involved in transmitting model updates between clients and the server, especially when dealing with models that have a large number of parameters.

Various strategies have been developed to mitigate the communication burden in FL. These techniques can be broadly classified into three categories: i) Compressing updates: sparsification [1], structured updates [3], and quantization [4, 5] reduce the size of transmitted model updates, ii) Local computation: performing multiple local epochs at the clients [6] lessens the frequency of communication with the server, and iii) Advanced aggregation methods and client selection: MOCHA [7] enhances the efficacy of update aggregation and [8, 9] reduce communication by only selecting a subset of clients to participate in each training round.

Existing FL techniques primarily rely on transmitting gradient signals or model updates, which are computed through backpropagation (BP). On the other hand, Evolutionary Strategies (ES) [10–12] update model parameters by utilizing fitness values obtained from evaluating a population of models. This approach eliminates the need for a gradient signal, as depicted in Fig. 1. Recent advances in neuroevolution have shown promise in supervised settings and competitive performance with reinforcement learning in control tasks [13–15, 11, 16–18]. In this context, ES offers a distinct advantage compared to traditional BP methods. This paradigm shift opens up the possibility of searching for novel solutions beyond the gradient-

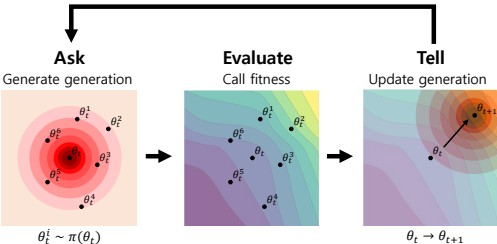

Figure 1: ES follows an iterative ask, evaluate, and tell approach for optimization. The strategy starts by generating candidate solutions of a base model (*ask*), which are then assessed using a fitness measure (*evaluate*). The base model is updated towards a better solution using the evaluation results (*tell*).

based methods. However, it is critical to note that ES, despite its potential, still lags behind gradient-based methods in certain problems like supervised learning. Fig. 2(a) reveals the performance gap between ES and BP. This emphasizes the need for a balanced approach that would leverage the strengths of both ES and gradient-based methods to achieve optimal results.

Our proposed method operates on the premise of incorporating high-quality gradient signals into the evaluation process of ES. The process can be formalized as follows:

Given a base model, denoted by $\theta$, we instantiate a population $P$ comprised of $N$ model samples. Each individual sample, $\theta^i$, is derived by adding random perturbations to $\theta$. Unlike traditional ES, where the fitness of $\theta^i$ corresponds to its performance on the task, we instead assess the fitness of each sample $\theta^i$ by measuring its similarity to $\theta'$, the model parameters updated through gradient descent steps. This operation effectively exploits the gradient signal to construct a fitness vector, a process we term Population-Based Gradient Encoding (PBGE). Fig. 2(b) shows the results of a representative experimental setting where PBGE is able to follow BP closely on the FMNIST dataset while maintaining an effective compression rate of over 98.8%. In particular, this method significantly outperforms sparsification strategies at equivalent compression rates.

In the context of FL, the gradient signal can be encoded into a fitness vector for the population of models and communicated to the server. For global synchronization, a well-established approach like FedAvg would involve reconstruction and aggregation of the clients' models and sharing of the aggregated model with the clients. However, by utilizing shared random seeds, we can ensure uniformity in the generated populations between clients and the server. This allows us

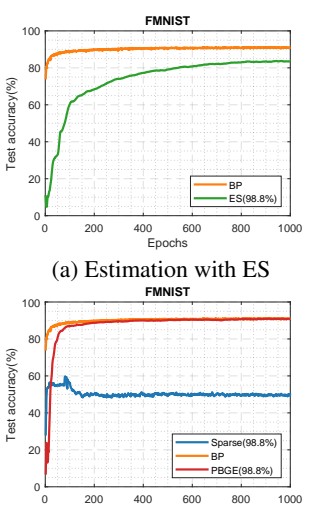

(a) Estimation with ES

(b) The proposed method

Figure 2: Test accuracy on FM-NIST dataset comparing (a) ES against BP, and (b) PBGE with BP and Sparse with 98.8% compression.

to transmit and aggregate only the small fitness vectors without reconstructing the models, reinforcing communication efficiency. In addition, there is an advantage in implementing extra privacy measures. When encryption is desired, the required overhead would be much smaller with the fitness vectors than with the model parameter vectors because of the size difference. A brief overview of EvoFed is shown in Fig. 3 (the detailed methodology is provided in Section 4).

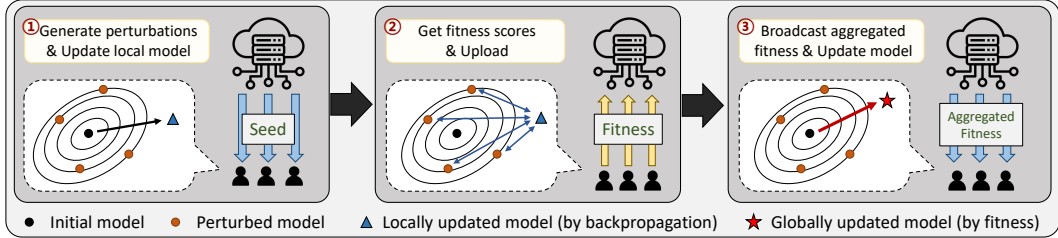

Figure 3: Overview of the proposed EvoFed: (1) Using the shared random seed, each client generates a population of perturbations around the local model. Each client also performs a gradient update of the local model using the local data as in conventional FL. (2) Each client evaluates the fitness of each perturbed model with respect to the updated model. The fitness values are communicated to the server. (3) The server aggregates the fitness values. Clients update their local models using broadcast aggregated fitness.

**Contributions.** To summarize, our main contribution is to introduce a novel concept of Population-Based Gradient Encoding (PBGE), which allows an accurate representation of the large local gradient vector using a relatively small fitness vector, significantly reducing the communication burden as well as the encryption overhead in FL environments. We also propose and verify the EvoFed framework that integrates PBGE into federated learning based on the exchange and aggregation of the fitness vectors between clients and the server.

Our approach achieves similar performance to FedAvg on both FMNIST and CIFAR-10 datasets. The advantage is that our scheme achieves very high compression, exceeding 98.8% on FMNIST and 99.7% on CIFAR-10 in key practical settings. Significantly, our EvoFed model also outperforms traditional compression methods, offering superior results at similar compression rates. The price we pay is the overhead of generating a population of perturbed models and computing similarity measures. This overhead could be substantial in the form of increased local processing time depending on the size of the population, but the client computational load can be traded with local memory space by employing parallel processing. In addition, as discussed later, reduce population size without affecting performance.

## 2 Related Work

**Federated Learning.** Several techniques have been proposed to alleviate the communication overhead in FL, such as model compression [1, 19, 20], distillation techniques [21–25], and client update subsampling [26, 27]. However, these approaches often come with trade-offs, including increased convergence time, lower accuracy, or additional computational requirements. More importantly, they still require the exchange of model parameters, which requires substantial communication bandwidth. Furthermore, [28] shows that the gradient sparsity of all participants has a negative impact on global convergence and communication complexity in FL.

**Evolutionary Strategies.** ES are black-box optimization algorithms inspired by biological evolution [29]. ES algorithms iteratively refine a population of solutions based on fitness evaluations. Natural Evolution Strategies (NES) is a specific variant of ES [30, 12, 31–35]. Within the NES framework, the distribution $p_\psi(\theta)$, parameterized by $\psi$, is adopted to represent the population and maximize the average objective value $\mathbb{E}_{\theta \sim p_\psi}[f(\theta)]$ via stochastic gradient ascent, where $f(\theta)$ is the fitness function. NES algorithms leverage a score function estimator as in [10]. Our method follows the guidelines provided by [36], where the parameter distribution $p_\psi$ is a factored Gaussian. Accordingly, we can represent $\mathbb{E}_{\theta \sim p_\psi}[f(\theta)]$ using the mean parameter vector $\theta$, such that $\mathbb{E}_{\theta \sim p_\psi}[f(\theta)] = \mathbb{E}_{\epsilon \sim \mathcal{N}(0,I)}[f(\theta + \sigma\epsilon)]$. Given a differentiable function estimator, the well-known conversion procedure allows optimization over $\theta$ to be rewritten as (see, for example, [37, 11, 38])

$$\nabla_\theta \mathbb{E}_{\epsilon \sim \mathcal{N}(0,I)}[f(\theta + \sigma\epsilon)] = \frac{1}{\sigma}\mathbb{E}_{\epsilon \sim \mathcal{N}(0,I)}\{f(\theta + \sigma\epsilon)\epsilon\}. \tag{1}$$

The strategy of sharing seeds for random number generators to synchronize parallel processes and maintain consistency is a well-established practice in areas ranging from parallel simulations to cryptographic protocols [39]. The appeal of shared random seeds stems from their ability to offer deterministic randomness, ensuring that multiple entities, operating independently, can produce synchronized and identical random outputs. Within the realm of ES, this concept was effectively harnessed by [36] to address the challenge of scalability. We also utilize the shared random seeds to

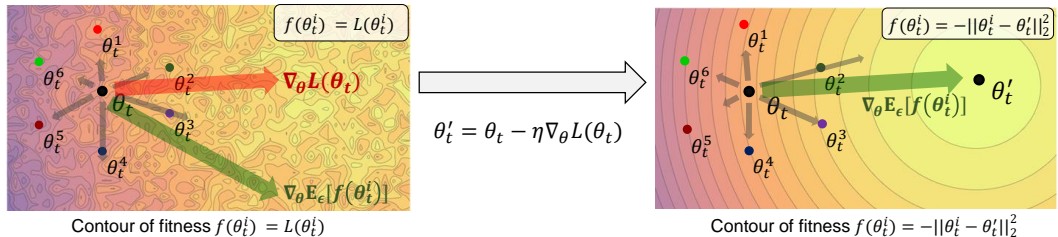

Figure 4: Illustration of one update step of typical ES and the proposed Population-Based Gradient Encoding (PBGE) strategy. **Left**: using loss value $L(\theta)$ directly for the fitness $f(\theta)$, as in existing ES, leads to an inherently noisy estimate of the gradient signal $\nabla_\theta \mathbb{E}_\epsilon[f(\theta + \sigma\epsilon)]$ due to the noisy loss surface. **Right**: PBGE defines fitness as the distance to the updated model $\theta'$ obtained through BP (i.e., $f(\theta) = -||\theta - \theta'||_2^2$). This enables the reliable estimate of $\theta'$ on the convex surface. By sampling a sufficient number of perturbations $\epsilon_i$, a decent gradient signal can be obtained, which aligns to the true gradient signal $\nabla_\theta L(\theta)$ computed from the BP.

scale down the communication load in a distributed setting. However, the key difference is that the work of [36] distributes the perturbations into multiple workers dealing with a single global objective, whereas our method generates identical perturbations across all clients, with each having a distinct objective. An important consequence is that with our method, each client only needs to communicate $N$ fitness values to update the model, instead of $MN$ values in [36] with $M$ denoting the number of nodes, enabling scalability regardless of the number of clients.

**Federated Learning and Evolutionary Strategies.** Several studies have explored the optimization of FL using ES. For instance, [40] introduced an evolutionary approach for network architecture search (NAS) in real-time FL, which minimizes local payload while optimizing model performance. Sparse evolutionary training (SET) [41] substitutes fully connected layers in neural networks with sparse layers to decrease the number of model parameters. Furthermore, [42] presented the SET algorithm that optimizes neural networks in FL via a bi-objective method to maximize accuracy performance while minimizing communication overhead. Additionally, [43] introduces the MOEA/D framework [44] to the environment of FL and FLEA [45] utilized Evolutionary Algorithms in FL setup at the client-level to evolve models.

While these studies have made significant contributions, the present work establishes a unique way of using ES as a method to reduce communication overhead in FL by transmitting fitness values instead of model parameters. In particular, the new gradient-driven fitness function essentially separates our work from the traditional utilization of ES for compression.

## 3   Population-Based Gradient Encoding

This work focuses on the process of encoding gradient information through an identical population of models generated at both ends of some network link. Population distribution is a zero mean isotropic multivariate Gaussian with fixed covariance $\sigma^2 I$. We also adopt 'mirrored sampling' [46, 47] for variance reduction where the Gaussian noise vector $\epsilon$ is instantiated with pairs of perturbations $\epsilon, -\epsilon$.

Given the reference point $\theta' = \theta - \eta\nabla L(\theta)$ where $\eta$ is the learning rate in the BP-based gradient update and $\nabla L(\theta)$ represents the gradient derived from the data, we define the fitness function $f(\theta)$ to measure the similarity between the model parameters $\theta$ and $\theta'$: $f(\theta) = -||\theta - \theta'||_2^2$. This choice ensures that the gradient of the expectation, $\nabla_\theta \mathbb{E}_{\epsilon \sim \mathcal{N}(0,I)}[f(\theta + \sigma\epsilon)]$, aligns with the actual gradient $\nabla L(\theta)$, effectively encoding the gradient information in the fitness values:

$$\nabla_\theta \mathbb{E}_{\epsilon \sim \mathcal{N}(0,I)}[-||(\theta + \sigma\epsilon) - \theta'||_2^2] = -\nabla_\theta||\theta - \theta'||_2^2 = 2(\theta - \theta') \tag{2}$$

where the first equality simply follows from the assumption that $\epsilon$ is zero-mean. The visual representation of this process is illustrated in Fig. 4. Eq. 2 gives $\theta' = \theta - \frac{1}{2}\nabla_\theta \mathbb{E}_\epsilon[-||(\theta + \sigma\epsilon) - \theta'||_2^2]$, and comparing with the BP operation $\theta' = \theta - \eta\nabla L(\theta)$, we have

$$\eta\nabla_\theta L(\theta) = \frac{1}{2}\nabla_\theta \mathbb{E}_{\epsilon \sim \mathcal{N}(0,I)}[-||(\theta + \sigma\epsilon) - \theta'||_2^2].$$

Now also utilizing Eq. 1, we write

$$\eta\nabla_\theta L(\theta) = \frac{1}{2\sigma}\mathbb{E}_{\epsilon \sim \mathcal{N}(0,I)}\{f(\theta + \sigma\epsilon)\epsilon\} \tag{3}$$

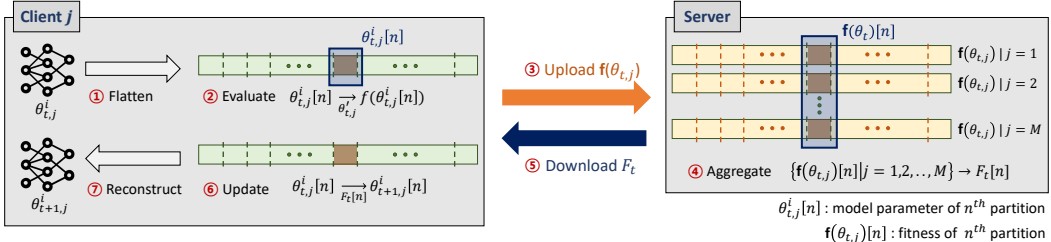

Figure 5: Partition-based Model Parameter Compression. Illustration of dividing model parameters $\theta$ into $K$ partitions and compressing them individually using PBGE.

where $f$ is the specific distance-based fitness function defined above. Approximating the expectation by sampling, we further write

$$\eta \nabla_\theta L(\theta) \approx \frac{1}{2N\sigma} \sum_{i=1}^{N} f(\theta + \sigma\epsilon_i)\epsilon_i \qquad (4)$$

which shows how the gradient update information can be encoded using the fitness values $f(\theta + \sigma\epsilon_i)$ corresponding to the perturbations $\epsilon_i$. Consequently, this removes the need to evaluate each perturbation with the dataset, as is done in existing ES, and the entire update process is encoded with low-cost distance measurement.

Finally, the update on $\theta$ itself based on the fitness values is naturally given by

$$\theta' \approx \theta + \frac{1}{2N\sigma} \sum_{i=1}^{N} f(\theta + \sigma\epsilon_i)\epsilon_i \qquad (5)$$

allowing a remote model update based on the transmitted $f(\theta + \sigma\epsilon_i)$ values with the shared knowledge of the $\epsilon_i$ values.

The implemented algorithm consistently executes three steps: (i) Compute the target $\theta'$ through gradient descent, (ii) Implement perturbations to the model parameters and assess the perturbed parameters by computing their Euclidean distance to $\theta'$, and (iii) Utilize the assessment results and encode the gradient with the fitness measures.

**Partitioning.** In the context of PBGE, the fitness function indicates the model distance, not the model performance on client data. This feature enables a unique partition-based approach for handling a large number of model parameters. Here, the parameters, flattened into a vector $\theta$, are split into $K$ partitions: $\theta[1], \theta[2], ..., \theta[K]$. Each partition is then effectively encoded individually using PBGE, which is advantageous when working with large models, as storing a large population of those may be burdensome for clients with limited memory. Partitioning allows us to compute $K$ fitness values per each perturbation, providing $K$ times more reference points to encode the gradient information given a population size. Hence, even with a small population size (requiring less memory), we achieve robust performance as validated by the empirical results provided in Supplementary Materials. Essentially, partitioning provides a tradeoff of memory with communication as more fitness values now need to be communicated per each perturbation. This partitioning process is visualized in Fig. 5. Notably, this partitioning technique can be used with any model architecture, regardless of its specific design, providing a practical and efficient means of compressing large models.

## 4 EvoFed

EvoFed operates on the principle that the evolutionary update step depends only on the fitness values given the perturbation samples. In the FL context, we can leverage this characteristic to devise an accurate yet communication-efficient strategy for model updates. This section provides a detailed exposition of our methodology, breaking it down into stages for clarity. This iterative process, as outlined in Algorithm 1, aims to gradually converge the model parameters at all nodes to an optimal solution while minimizing data transmission during each update. An overall view of our proposed methodology is depicted in Fig. 3.

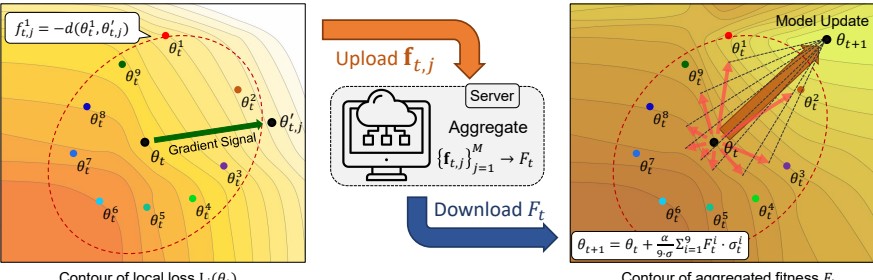

Figure 6: Local Updates and Fitness Evaluation (left) and Server/Client-side Update (right). Left: Client performs BP on local data $\theta_t$ to obtain $\theta'_{t,j}$, evaluates fitness $f^i_{t,j}$ by distance measure (e.g. L2) with $\theta^i_t$, and uploads $\mathbf{f}_{t,j}$ to the server. Right: After obtaining the aggregated fitness $\mathbf{F}_t$, all nodes update the baseline model $\theta_t$ according to Eq. 9.

## 4.1 Initial Setup

The initialization of the server and clients in EvoFed begins with the same baseline model, denoted by $\theta_0$. A key assumption is that the server and all clients share the same seed (e.g., via broadcasting) for identical random population generation. This approach ensures consistent baseline models and populations across all nodes.

In this step, a population of candidate solutions (in this case, models) is generated by adding random perturbations to the current best solution (the baseline model). The generation of an $i$-th member of the model population can be formulated as follows:

$$\theta^i_t = \theta_t + \mathcal{N}(0, \sigma I) \tag{6}$$

Here, $\mathcal{N}(0, \sigma I)$ represents the perturbations sampled from a multivariate normal distribution with zero mean and a shared covariance matrix $\sigma I$. We also denote the population at each node at any given time $t$ as $\mathbf{P}_t = \{\theta^1_t, \theta^2_t, ..., \theta^N_t\}$, where $\theta^i_t$ represents the $i$-th model in the population generated by adding the $i$-th perturbation to the baseline model parameters, and $N$ is the size of the population.

## 4.2 Local Updates and Fitness Evaluation

Each client node begins by executing BP on its local dataset, using the baseline model, $\theta_t$, resulting in an updated model, $\theta'_t$. Following this, the fitness of each member $\theta^i_t$ in the local population, $\mathbf{P}_t$, is evaluated. This evaluation is done by measuring the similarity between $\theta'_t$ and $\theta^i_t$. The L2 norm or Euclidean distance serves as the measure of this similarity. The fitness of $\theta^i_t$ is represented as $f(\theta^i_t)$:

$$f(\theta^i_t) = -||\theta'_t - \theta^i_t||^2_2 \tag{7}$$

The process of local update and fitness evaluation is illustrated in Fig. 6. The fitness values are the only information that needs to be communicated among nodes. The fitness vectors are significantly smaller in size compared to the model parameter vectors, which helps reduce the communication overhead. Hence, each client sends a fitness vector, $\mathbf{f}_t = \{f(\theta^1_t), f(\theta^2_t), ..., f(\theta^N_t)\}$ corresponding to all population members, to the server.

## 4.3 Server-side Aggregation and Update

The server's responsibility is to aggregate the fitness values reported by all client nodes, forming a global fitness vector $\mathbf{F}_t$ comprising $N$ elements, with $\mathbf{F}^i_t$ representing the fitness value of the $i^{th}$ member of the population. Each client's contribution to $\mathbf{F}_t$ is weighted by their respective batch size $b_j$, giving a larger influence to clients with potentially more accurate local updates.

The global fitness vector $\mathbf{F}_t$ is computed as follows:

$$\mathbf{F}_t = \frac{1}{\sum_{j=1}^M b_j} \sum_{j=1}^M b_j \mathbf{f}_{t,j} \tag{8}$$

where $M$ is the total number of clients and $\mathbf{f}_{t,j}$ is the fitness value vector from client $j$ at time $t$. After the aggregation, the server broadcasts $\mathbf{F}_t$ so that each client updates the baseline model $\theta_t$.

---

**Algorithm 1** EvoFed: Federated Learning with Evolutionary Strategies

---
1: **Input:** Learning rates $\eta, \alpha$, population size $N$, noise std $\sigma$, seed $s$
2: **Initialize:** server and $M$ clients with seed $s$ and identical parameters $\theta_0$ and $\theta_{0,j}$ respectively
3: **for** each communication round $t = 0, 1, ..., T - 1$ in parallel **do**
4:     **for** each client $j$ in parallel **do**
5:         $\theta'_{t,j} = \theta_{t,j} - \eta\nabla_\theta L(\theta'_{t,j})$                 //Backpropagation Update
6:         Sample $\epsilon_t^i \sim \mathcal{N}(0, \sigma I)$                 //Sample perturbations
7:         $\theta_{t,j}^i = \theta_{t,j} + \epsilon_t^i$ for $i = 1, ..., N$            //Initialize population
8:         $\mathbf{f}_{t,j}^i = -\|\theta_{t,j}^i - \theta'_{t,j}\|_2^2$ for $i = 1, ..., N$     // Compute fitness vector using $L_2$ calculation
9:     **end for**
10:    $\mathbf{F}_t^i = \frac{1}{\sum_j^M b_j}\sum_j^M b_j\mathbf{f}_{t,j}^i$ for $i = 1, ..., N$       // Server averages fitness vectors
11:    $\theta_{t+1} = \theta_t + \frac{\alpha}{N\sigma}\sum_{i=1}^N \mathbf{F}_t^i \cdot \epsilon_t^i$       // Server updates model using the aggregated fitness
12:    **Broadcast** $\{\mathbf{F}_t^1, \mathbf{F}_t^2, ..., \mathbf{F}_t^N\}$
13:    **for** each client $j$ in parallel **do**
14:         $\theta_{t+1,j} = \theta_{t,j} + \frac{\alpha}{N\sigma}\sum_{i=1}^N \mathbf{F}_t^i \cdot \epsilon_t^i$     // Client updates model using the aggregated fitness
15:    **end for**
16: **end for**

---

## 4.4 Broadcasting Fitness

Once the aggregation is complete, the server broadcasts $\mathbf{F}_t$ to all client nodes, maintaining synchronicity across the network. This only involves the transmission of the fitness vector, again reducing communication overhead. The aggregated fitness vector can be used by the local clients (as well as by the server, if needed) to update the local models.

## 4.5 Client-side Update

When the global fitness vector $\mathbf{F}_t$ is received from the server, the clients can update their local model following Eq. 5. This strategy involves the addition of a weighted sum of noise vectors to the current model parameters, where the weights are aggregated fitness values:

$$\theta_{t+1} = \theta_t + \frac{\alpha}{N\sigma}\sum_{i=1}^N \mathbf{F}_t^i \cdot \epsilon_t^i \tag{9}$$

where $\epsilon_t^i$ is the $i$-th noise vector from the population at time $t$.

Notably, the right side of Eq. 9 can be shown equivalent to the average of all locally updated models akin to FedAvg, i.e., it is straightforward to show that

$$\theta_{t+1} = \frac{1}{\sum_j^M b_j}\sum_{j=1}^M b_j\theta_{t+1,j} \tag{10}$$

where $\theta_{t+1,j}$ is the updated model of client $j$ at time $t+1$ defined as

$$\theta_{t+1,j} = \theta_t + \frac{\alpha}{N\sigma}\sum_{i=1}^N f_{t,j}^i \cdot \epsilon_t^i. \tag{11}$$

## 4.6 Convergence Analysis

**Theorem 1.** *Suppose that $L_j(\theta)$ is the $\beta$-smooth function, i.e., $\|\nabla L_j(u) - \nabla L_j(v)\| \leq \beta\|u - v\|$ for any $u, v$, and also suppose that the variance of the stochastic gradient of $\tilde{D}_j$ is bounded, i.e., $\mathbb{E}\|\nabla L_j(\theta) - \widetilde{\nabla} L_j(\theta)\|^2 \leq B^2$ for all $j$. When perturbation $\epsilon^i$ is sampled, a conditioned mirrored sampling is applied such that $\frac{1}{N}\sum_{i=1}^N \epsilon^i = 0$, $\frac{1}{N}\sum_{i=1}^N (\epsilon^i)^2 \leq G^2$, $\frac{1}{N}\sum_{i=1}^N (\epsilon^i)^3 = 0$. Given a decreasing learning rate $\eta_t < \frac{1}{4\alpha\beta}$, EvoFed converges in the sense of*

$$\frac{1}{H_T}\sum_{t=0}^{T-1}\eta_t\mathbb{E}\left[\|\nabla L(\theta_t)\|^2\right] \leq \frac{\mathbb{E}\left[L(\theta_0)\right] - L^*}{\alpha G^2 H_T} + 4\alpha\beta B^2\left(\frac{1}{H_T}\sum_{t=0}^{T-1}\eta_t^2\right)$$

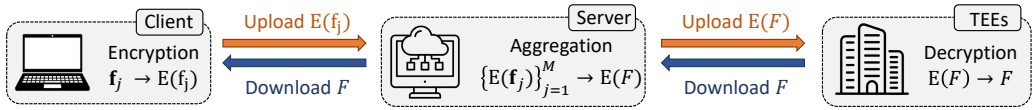

Figure 7: Privacy enhancement through Fully Homomorphic Encryption and Trusted Execution Environments.

*where $H_T = \sum_{t=0}^{T-1} \eta_t$, and $L^*$ represents the minimum value of $L(\theta)$.*

Given a decreasing learning rate (e.g., $\eta_t = \frac{\eta_0}{1+t}$), it can be seen that $H_T = \sum_{t=0}^{T-1} \eta_t \to \infty$ as $T$ increases, while $\sum_{t=0}^{T-1} \eta_t^2 < \infty$. Consequently, the upper bound stated in Theorem 1 approaches 0 as $T$ grows, ensuring convergence towards a stationary point. The detailed discussions and the proof can be found in Supplementary Materials.

### 4.7 EvoFed and Privacy Enhancements

Unlike other FL methods, EvoFed operates on fitness measures. Encryption on smaller fitness vectors requires a lower overhead compared to encryption on large model parameter vectors. Fig. 7 shows Fully Homomorphic Encryption (FHE) [48–50] that allows aggregation of encrypted data on the server while keeping individual client data confidential. For the decryption of the aggregated fitness vector, EvoFed can leverage third-party Trusted Execution Environments (TEEs), such as Intel SGX [51], providing a secure space for sensitive computations.

### 4.8 Partial Client Participation

When FL has to operate with a large client pool, a typical practice is to select only a subset of clients in each global round. In EvoFed, this means that the newly joined clients in a given round can either download the latest model or else the last $k$ fitness vectors from the server, where $k$ is the time gap from the last participation. In the latter case, the model updates from Eq. (9) can be modified to accommodate $k$-step updates:

$$\theta_t = \theta_{t-k} + \frac{\alpha}{N\sigma} \sum_{l=1}^{k} \sum_{i=1}^{N} \mathbf{F}_{t-l}^i \cdot \epsilon_{t-l}^i. \tag{12}$$

Note that even in the former case where the latest model is downloaded by the newly joined clients, the level of security is not compromised as far as the client-to-server messages are concerned.

## 5 Experiments

Our algorithm's effectiveness is assessed on three image classification datasets: FMNIST [52], MNIST [53], and CIFAR-10 [54]. Both MNIST and FMNIST contain 60,000 training samples and 10,000 test samples, whereas CIFAR-10 is composed of 50,000 training samples and 10,000 test samples. We employ a CNN model having 11k parameters for the MNIST and FMNIST datasets and a more substantial model with 2.3M parameters for CIFAR-10. A grid search has been conducted to identify the optimal performance hyperparameters for each baseline, as outlined in the results section. We take into account both global accuracy and communication costs to ascertain hyperparameters that maximize accuracy while minimizing communication overhead.

**Data Distribution.** We distribute the training set of each dataset among clients for model training, and the performance of the final global model is evaluated using the original test set. Our experimental setup contains $M = 5$ clients with non-IID data distribution (assigning two classes to each client).

**Implementation Details.** Our EvoFed framework is built using JAX [55], which facilitates extensive parallelization and, in particular, consistent random number generation across a large number of nodes. We have implemented our framework on the Evosax [56] library, a convenient tool for the ES algorithm. EvoFed is configured with a population size of 128 and a mini-batch size of 256 for MNIST / FMNIST and 64 for CIFAR-10. We perform ten local epochs (performing ten BP steps before fitness calculation) and train over 1,000 global rounds.

**Baselines.** We compare the performance of the proposed EvoFed with BP, FedAvg, ES, FedAvg with quantization (Fed-quant), and FedAvg with Sparsification (Fed-sparse). In each scenario, we push

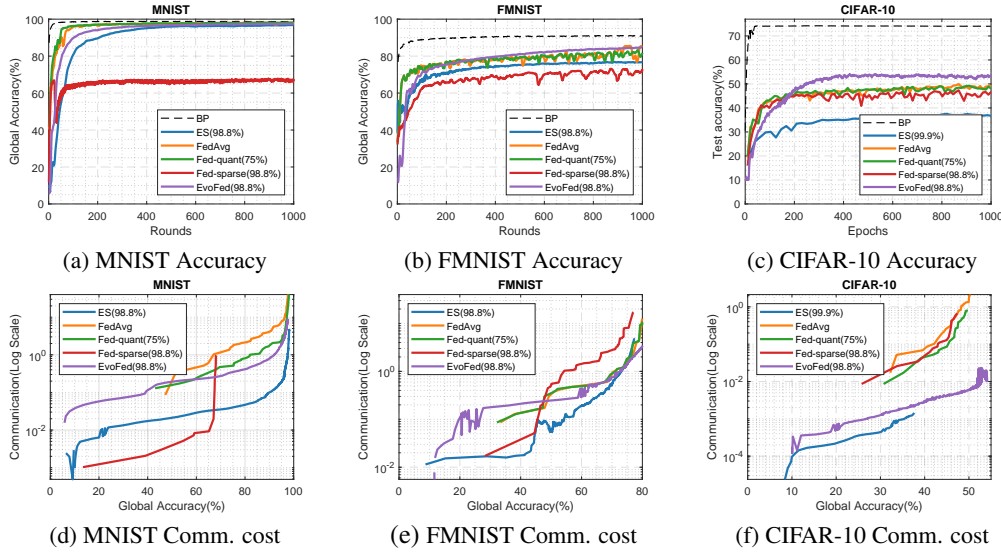

Figure 8: Performance comparison of EvoFed and baseline methods on MNIST, FMNIST, and CIFAR-10 datasets. The top row displays the accuracy achieved by each method on the respective datasets, while the bottom row illustrates the communication cost associated with each method.

for maximum compression, stopping right before the model starts to show performance degradation relative to FedAvg with no compression. BP provides the upper-performance baseline, while ES serves as a reference emphasizing the significance of PBGE.

## 6 Results and Discussions

In this section, we discuss the experimental results in detail and provide further insights into the performance of EvoFed. The accuracy of EvoFed, compared with multiple baseline methods and different datasets, is shown in Fig. 8 (a), (b), and (c). Efficiently encoding and exchanging gradient information, EvoFed enhances the effectiveness of the ES algorithm across all tasks, delivering results comparable to FedAvg. Also, EvoFed achieves superior accuracy at an equivalent compression rate compared to sparsification. This suggests that utilizing a shared population of samples can reduce the information necessary for gradient compression, thereby enhancing the efficiency of the process. Fig. 8 (d), (e), and (f) shows the performance of each method as a function of communication load for all three datasets. It can be seen that EvoFed tends to utilize significantly less communication resources as compared to other high-accuracy techniques.

Table 1 summarizes the performance of different schemes on MNIST, FMNIST, and CIFAR-10 datasets, focusing on communication cost and accuracy. EvoFed achieves significantly lower communication costs compared to FedAvg while maintaining competitive accuracy levels. In the MNIST dataset, EvoFed achieves an accuracy of 97.62% with a mere 9.2 MB of communication load, while FedAvg achieves 98.09% accuracy at a considerably high communication cost of 73.7 MB. The effective compression achieved is an impressive 98.8% which indicates that the gradient vector is condensed into just 1.2% of the fitness vector that is communicated between clients and the server. Similarly, for the FMNIST dataset, EvoFed achieves an accuracy of 84.72% with only 7.78 MB of communication, whereas FedAvg's accuracy is 85.53% with a communication cost of 40.99 MB. The efficiency of EvoFed becomes even more apparent in the CIFAR-10 dataset where the model compression is over 99.7%. EvoFed achieves an accuracy of 54.12% with a low communication cost of only 0.023 GB, surpassing FedAvg, which performs 50.22% at a communication cost of 2.134 GB. The simpler ES method actually gives better performance as well as higher communication efficiency than EvoFed for MNIST but its accuracy for other data sets is highly limited.

**Additional Experiments**. Fig. 9 illustrates the performance of EvoFed with varying numbers of samples $N$ within the population and a varying number of clients $M$. The model's performance is

| Methods | MNIST | | | FMNIST | | | CIFAR-10 | | |
|---|---|---|---|---|---|---|---|---|---|
| | Comm. Cost (MB) | | Max | Comm. Cost (MB) | | Max | Comm. Cost (GB) | | Max |
| | 90% Acc. | Max Acc. | Acc. | 70% Acc. | Max Acc. | Acc. | 45% Acc. | Max Acc. | Acc. |
| ES | 0.1 | 4.6 | 98.30% | 0.48 | 4.85 | 76.86% | - | 0.021 | 37.47% |
| FedAvg | 4.2 | 73.7 | 98.09% | 0.87 | 40.99 | 85.53% | 0.266 | 2.134 | 50.22% |
| Fed-quant | 1.7 | 41.8 | 98.15% | 0.94 | 37.98 | 83.23% | 0.086 | 0.800 | 49.78% |
| Fed-sparse | - | 1.0 | 67.85% | 2.78 | 17.13 | 73.17% | 0.129 | 0.671 | 47.36% |
| EvoFed (our) | 0.8 | 9.2 | 97.62% | 0.75 | 7.78 | 84.72% | 0.004 | 0.023 | 54.12% |

Table 1: Performance of different schemes presented in tabular form, corresponding to Fig. 8.

significantly influenced by the population size, which directly affects the algorithm's exploration capability and compression rate. We observe a performance improvement as we increase the population size, although at the cost of increased computation and memory usage. Nonetheless, this improvement is not linear and plateaus once sufficient samples are generated to explore the parameter space. For instance, in the case of the FMNIST dataset and a model with 11K parameters, any performance enhancement beyond a generated sample size of 128 is marginal.

Fig. 9(b) showcases EvoFed's performance trend as the number of clients increases. A minor performance decline is observable for some larger values of M relative to M=5. This decline could potentially be attributed to the limited data available per client and the increased variance associated with local training. Nonetheless, we believe that carefully tuning the hyperparameters based on data distribution could assist in achieving more robust performance.

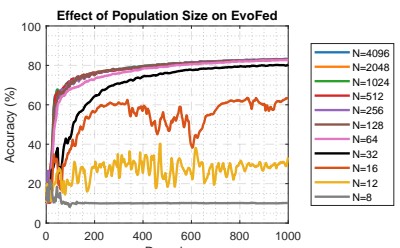 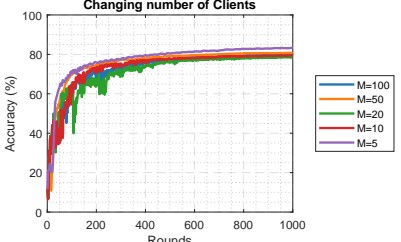

(a) Varying number of population from 8 to 4096    (b) Varying number of clients from 5 to 100

Figure 9: Effect of population size (left) and number of clients (right) on EvoFed

Supplementary Materials provide further details regarding hyperparameters, model architectures, and experiments. We also include a detailed ablation study with different evolutionary strategies, the population size, and the number of partitions, as well as detailed communication and computational complexity analyses.

## 7 Conclusion

EvoFed, the novel paradigm presented herein, unites FL and ES to offer an efficient decentralized machine learning strategy. By exchanging compact fitness values instead of extensive model parameters, EvoFed significantly curtails communication costs. Its performance parallels FedAvg while demonstrating an impressive equivalent model compression of over 98.8% on FMNIST and 99.7% on CIFAR-10 in representative experimental settings. Consequently, EvoFed represents a substantial stride forward in FL, successfully tackling the critical issue of high communication overhead.

## Acknowledgments

A special acknowledgment goes to Do-yeon Kim for the valuable discussions on the convergence analysis presented in Section 4. This work was supported by the National Research Foundation of Korea (NRF) grant funded by the Korean government (MSIT) (No. NRF-2019R1I1A2A02061135), and by IITP funds from MSIT of Korea (No. 2020-0-00626).

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
