## Supplementary Materials

## A  Complexity Analysis

Our proposed method significantly reduces communication overhead in federated learning. However, this reduction in communication comes at the cost of an increase in computation and memory usage on the client side. In this section, we provide a comprehensive analysis of these complexity trade-offs and discuss potential strategies to alleviate the added burdens. Specifically, we present an analysis of the computational, memory, and communication complexities of our proposed model and provide a comparative assessment against existing baselines.

In this analysis, $E$ stands for the number of local epochs executed within a single communication round, $M$ indicates the total number of clients, $N$ represents the population size per client, and $|\theta|$ denotes the dimension of the model parameter vector. Table 1 presents the order of time and memory complexities for each method where we do not employ parallelization for ES and EvoFed computation. Here, clients generate perturbations one by one and reuse the memory after the fitness measurement of a perturbed sample. As shown in Table 1, EvoFed without parallelization has a similar order of memory complexity in clients to conventional FL, i.e., FedAvg, and reduced memory complexity in the server. However, in this scenario, the time complexity for EvoFed grows linearly with the number of perturbations as compared to FedAvg.

Table 1: Comparison of time and memory complexities for ES, EvoFed, and FedAvg, without parallel processing of $N$ individual perturbed models.

| Method | Client Time | Client Memory | Server Time | Server Memory |
|---|---|---|---|---|
| ES | $O(N|\theta|)$ | $O(|\theta|)$ | $O(N(|\theta| + M))$ | $O(|\theta| + MN)$ |
| EvoFed | $O(N|\theta| + E|\theta|)$ | $O(|\theta|)$ | $O(N(|\theta| + M))$ | $O(|\theta| + MN)$ |
| FedAvg | $O(E|\theta|)$ | $O(|\theta|)$ | $O(M|\theta|)$ | $O(M|\theta|)$ |

To mitigate the time complexity in EvoFed, one approach is to generate and evaluate a batch of $T$ perturbations in parallel. This method poses a trade-off between time and memory complexity.

Partitioning (as discussed in the main text) is an alternative strategy that enables computing a higher number of fitness values for each perturbation by dividing it into $K$ partitions. Consequently, the algorithm requires a fewer perturbations $N'$ to obtain sufficient fitness values for gradient encoding, resulting in a reduction in memory complexity. This necessitates the transmission of $N'K$ fitness values to the server. While partitioning does not introduce additional time complexity, having a small number of perturbations restricts the algorithm's exploration capabilities in parameter space, as discussed in Section D.3. Therefore, in practice, we choose the population size to be $\frac{N}{K} \leq N' \leq N$, trading memory complexity with communication cost.

Table 2 provides a comparative analysis of the time and memory complexities for EvoFed and FedAvg when both $T$ individual perturbed models are processed in parallel, and each perturbed model sample is divided into $K$ partitions. In scenarios with enough memory, it is feasible to execute all perturbations in a parallel setting $T = N$ and without partitioning $K = 1$ and $N = N'$.

Table 2: Comparison of time and memory complexities for EvoFed and FedAvg, with parallel processing of $T$ individual perturbed models and where each perturbation is partitioned to $K$ segments.

| Method | Client Time | Client Memory | Server Time | Server Memory |
|---|---|---|---|---|
| EvoFed (Parallel) | $O(\frac{N}{T}|\theta| + E|\theta|)$ | $O(T|\theta|)$ | $O(\frac{N}{T}(|\theta| + M))$ | $O(N|\theta| + MN)$ |
| EvoFed (Partitioned) | $O(N'|\theta| + E|\theta|)$ | $O(N'|\theta|)$ | $O(N'(|\theta| + M))$ | $O(N'|\theta| + MKN')$ |
| EvoFed (Both) | $O(\frac{N'}{T}|\theta| + E|\theta|)$ | $O(T|\theta|)$ | $O(\frac{N'}{T}(|\theta| + M))$ | $O(N'|\theta| + MKN')$ |
| FedAvg | $O(E|\theta|)$ | $O(|\theta|)$ | $O(M|\theta|)$ | $O(M|\theta|)$ |

As discussed before, the communication complexity of ES and EvoFed is limited to transferring fitness values $O(N'K)$ while FedAvg is to the gradient signal $O(|\theta|)$. However we can apply additional compression on each method to reduce this complexity as explored in D.1 and D.2.

## B  Model Architecture and Optimization Hyperparameters

We used a CNN model with 11k parameters for the MNIST and FMNIST datasets and a bigger model with 2.3M parameters for CIFAR-10, with architectural details provided in Table 5 and Table

6 respectively. We also provide detailed information about the optimization hyperparameters e.g. learning rate (lr), momentum and batch size, etc. for MNIST and FMNIST in Table 3 and for Cifar-10 in Table 4:

Table 3: Hyperparameters used in experiments on dataset MNIST & FMNIST

| Model | Methods | Hyperparameters | | | | | | | | | | |
|-------|---------|-----------------|------|----------|-----------|--------|-------------|-------------|---------|-------|-----|-----------|
| | | batch size | lr | momentum | optimizer | lr_es | momentum_es | optimizer_es | w_decay | sigma | eps | $\beta\_1\&\beta\_2$ |
| CNN | ES | 128 | - | - | - | 0.0148 | 0.9 | sgd | 0.0 | 0.27 | 1e-8 | 0.99 & 0.999 |
| | FedAvg | 256 | 0.0111 | 0.8099 | sgd | - | - | - | - | - | - | - |
| | Fed-quant | 256 | 0.0111 | 0.8099 | sgd | - | - | - | - | - | - | - |
| | Fed-sparse | 256 | 0.0111 | 0.8099 | sgd | - | - | - | - | - | - | - |
| | **EvoFed (ours)** | 256 | 0.0873 | 0.9074 | sgd | 0.0427 | 0.9 | sgd | 0.0152 | 0.27 | 1e-8 | 0.99 & 0.999 |

Table 4: Hyperparameters used in experiments on dataset CIFAR-10

| Model | Methods | Hyperparameters | | | | | | | | | | |
|-------|---------|-----------------|------|----------|-----------|--------|-------------|-------------|---------|-------|-----|-----------|
| | | batch size | lr | momentum | optimizer | lr_es | momentum_es | optimizer_es | w_decay | sigma | eps | $\beta\_1\&\beta\_2$ |
| CNN | ES | 32 | - | - | - | 0.04 | 0.4815 | sgd | 0.0 | 0.35 | 1e-8 | 0.99 & 0.999 |
| | FedAvg | 128 | 0.0009 | 0.6132 | sgd | - | - | - | - | - | - | - |
| | Fed-quant | 128 | 0.0009 | 0.6132 | sgd | - | - | - | - | - | - | - |
| | Fed-sparse | 128 | 0.0009 | 0.6132 | sgd | - | - | - | - | - | - | - |
| | **EvoFed (ours)** | 64 | 0.0148 | 0.3011 | sgd | 0.0275 | 0.5239 | sgd | 0.0824 | 0.35 | 1e-8 | 0.99 & 0.999 |

Table 5: Detailed information of the CNN architecture used in MNIST & FMNIST experiments

| Layer | Parameter & Shape (cin, cout, kernal size) & hyper-parameters | # |
|-------|----------------------------------------------------------------|---|
| layer1 | conv1: $1 \times 8 \times 5 \times 5$, stride:(1, 1); padding:0 | $\times 1$ |
| | avgpool | $\times 1$ |
| layer2 | conv1: $8 \times 16 \times 5 \times 5$, stride:(1, 1); padding:0 | $\times 1$ |
| | avgpool | $\times 1$ |
| | fc: $16 \times 10$ | $\times 1$ |

Table 6: Detailed information of the CNN architecture used in CIFAR-10 experiments

| Layer | Parameter & Shape (cin, cout, kernal size) & hyper-parameters | # |
|-------|----------------------------------------------------------------|---|
| layer1 | conv1: $3 \times 64 \times 5 \times 5$, stride:(1, 1); padding:0 | $\times 1$ |
| | avgpool | $\times 1$ |
| layer2 | conv1: $64 \times 128 \times 5 \times 5$, stride:(1, 1); padding:0 | $\times 1$ |
| | avgpool | $\times 1$ |
| | fc: $128 \times 256$ | $\times 1$ |
| | fc: $256 \times 10$ | $\times 1$ |

## C Convergence Analysis

**Assumption 1.** *For each $j$, $L_j(v)$ is $\beta$-smooth, i.e., $\|\nabla L_j(u) - \nabla L_j(v)\| \leq \beta \|u - v\|$ for any $u, v$.*

**Assumption 2.** *Variance of the gradient of $D_j$ is bounded, $\mathbb{E}\left[\left\|\nabla L_j(\theta) - \widetilde{\nabla} L_j(\theta)\right\|^2\right] \leq B^2$.*

**Assumption 3.** *When perturbation $\epsilon^i$ is sampled from the population distribution $p_\psi$, a conditioned mirrored sampling is applied such that $\frac{1}{N}\sum_{i=1}^N \epsilon^i = 0$, $\frac{1}{M}\sum_{i=1}^N \left(\epsilon^i\right)^2 \leq G^2$, $\frac{1}{N}\sum_{i=1}^N \left(\epsilon^i\right)^3 = 0$.*

**Theorem 1.** *Given a decreasing learning rate $\eta_t < \frac{1}{4\alpha\beta}$, EvoFed has the convergence bound as:*

$$\frac{1}{H_T}\sum_{t=0}^{T-1}\eta_t\mathbb{E}\left[\|\nabla L(\theta_t)\|^2\right] \leq \frac{\mathbb{E}\left[L(\theta_0)\right] - L^*}{\alpha G^2 H_T} + 4\alpha\beta B^2\left(\frac{1}{H_T}\sum_{t=0}^{T-1}\eta_t^2\right)$$

*where $H_T = \sum_{t=0}^{T-1}\eta_t$, and $L^*$ represents the minimum value of $L(\theta)$.*

By $\beta$-smoothness of $L(\theta)$ and taking expectation on both sides, we have

$$\mathbb{E}\left[L(\theta_{t+1}) - L(\theta_t)\right] \leq \mathbb{E}\left[\langle \nabla L(\theta_t), \theta_{t+1} - \theta_t \rangle\right] + \frac{\beta}{2}\mathbb{E}\left[\|\theta_{t+1} - \theta_t\|^2\right] \tag{1}$$

**Proof.** By utilizing the proof of Lemma 1 and recognizing $\langle \cdot, \cdot \rangle$ as the inner product operation, we rewrite the first term $\mathbb{E}\left[\langle \nabla L(\theta_t), \theta_{t+1} - \theta_t \rangle\right]$ as follows:

$$\mathbb{E}\left[\langle \nabla L(\theta_t), \theta_{t+1} - \theta_t \rangle\right] \stackrel{(a)}{=} \mathbb{E}\left[\left\langle \nabla L(\theta_t), \frac{1}{M}\sum_{j=1}^{M}\frac{\alpha}{N\sigma}\sum_{i=1}^{N} f(\theta_{t,j} + \sigma\epsilon_t^i)\epsilon_t^i \right\rangle\right]$$

$$\stackrel{(b)}{=} \mathbb{E}\left[\left\langle \nabla L(\theta_t), \frac{1}{M}\sum_{j=1}^{M}\frac{\alpha}{N\sigma}\sum_{i=1}^{N} \|(\theta_{t,j} + \sigma\epsilon_t^i) - (\theta_{t,j}')^i\|^2\epsilon_t^i \right\rangle\right]$$

$$= -\mathbb{E}\left[\left\langle \nabla L(\theta_t), \frac{1}{M}\sum_{j=1}^{M}\frac{\alpha}{N\sigma}\sum_{i=1}^{N} \|(\theta_{t,j} + \sigma\epsilon_t^i)\right.\right.$$

$$\left.\left. - (\theta_{t,j} - \eta_t\widetilde{\nabla}L_j(\theta_{t,j}))\|^2\epsilon_t^i \right\rangle\right]$$

$$= -\mathbb{E}\left[\left\langle \nabla L(\theta_t), \frac{1}{M}\sum_{j=1}^{M}\frac{\alpha}{N\sigma}\sum_{i=1}^{N} \|(\sigma\epsilon_t^i) + \eta_t\widetilde{\nabla}L_j(\theta_{t,j})\|^2\epsilon_t^i \right\rangle\right]$$

$$= -\mathbb{E}\left[\left\langle \nabla L(\theta_t), \frac{1}{M}\sum_{j=1}^{M}\frac{\alpha}{N\sigma}\sum_{i=1}^{N} \left(\sigma^2\left(\epsilon_t^i\right)^3 + 2\sigma\left(\epsilon_t^i\right)^2\eta_t\widetilde{\nabla}L_j(\theta_t)\right.\right.\right.$$

$$\left.\left.\left. + \epsilon_t^i\eta_t^2\|\widetilde{\nabla}L_j(\theta_t)\|^2\right) \right\rangle\right]$$

$$\stackrel{(c)}{\leq} -\mathbb{E}\left[\left\langle \nabla L(\theta_t), \frac{1}{M}\sum_{i=j}^{M}\frac{\alpha}{\sigma}\left(2\sigma G^2\eta_t\widetilde{\nabla}L_j(\theta_t)\right) \right\rangle\right]$$

$$\stackrel{(d)}{=} (-2\alpha\eta_t G^2)\,\mathbb{E}\left[\left\langle \nabla L(\theta_t), \frac{1}{M}\sum_{j=1}^{M}\nabla L_j(\theta_t) \right\rangle\right]$$

$$\stackrel{(e)}{=} (-\alpha\eta_t G^2)\left\{\mathbb{E}\left[\|\nabla L(\theta_t)\|^2\right] + \mathbb{E}\left[\|\frac{1}{M}\sum_{j=1}^{M}\nabla L_j(\theta_t)\|^2\right]\right.$$

$$\left. - \underbrace{\mathbb{E}\left[\|\nabla L(\theta_t) - \frac{1}{M}\sum_{j=1}^{M}\nabla L_j(\theta_t)\|^2\right]}_{=0}\right\}$$

where $(a)$ comes from the Lemma 1, $(b)$ is due to $f(\theta_t) = -\|\theta_t - \theta_t'\|^2$, $(c)$ follows from Assumption 3, $(d)$ is from taking expectation for the mini-batch, and $(e)$ is due to the well-known equality $\|z_1 - z_2\|^2 = \|z_1\|^2 + \|z_2\|^2 - 2\langle z_1, z_2 \rangle$.

On the other hand, we can bound the second term $\mathbb{E}\left[\|\theta_{t+1} - \theta_t\|^2\right]$ as follows:

$$\mathbb{E}\left[\|\theta_{t+1} - \theta_t\|^2\right] = \mathbb{E}\left[\left\|\frac{1}{M}\sum_{j=1}^{M}\frac{\alpha}{N\sigma}\sum_{i=1}^{N}f(\theta_t + \sigma\epsilon_t^i)\epsilon_t^i\right\|^2\right]$$

$$= \mathbb{E}\left[\left\|\frac{1}{M}\sum_{j=1}^{M}\frac{\alpha}{N\sigma}\sum_{i=1}^{N}\|(\theta_t + \sigma\epsilon_t^i) - \theta_t'\|^2\epsilon_t^i\right\|^2\right]$$

$$= \mathbb{E}\left[\left\|\frac{1}{M}\sum_{j=1}^{M}\frac{\alpha}{N\sigma}\sum_{i=1}^{N}\|(\theta_t + \sigma\epsilon_t^i) - (\theta_t - \eta_t\widetilde{\nabla}L_j(\theta_t))\|^2\epsilon_t^i\right\|^2\right]$$

$$= \mathbb{E}\left[\left\|\frac{1}{M}\sum_{j=1}^{M}\frac{\alpha}{N\sigma}\sum_{i=1}^{N}\left(\sigma^2\left(\epsilon_t^i\right)^3 + 2\sigma\left(\epsilon_t^i\right)^2\eta_t\widetilde{\nabla}L_j(\theta_t) + \epsilon_t^i\eta_t^2\|\widetilde{\nabla}L_j(\theta_t)\|^2\right)\right\|^2\right]$$

$$\underset{(a)}{\leq}\mathbb{E}\left[\left\|\frac{1}{M}\sum_{j=1}^{M}\frac{\alpha}{\sigma}\left(2\sigma G^2\eta_t\widetilde{\nabla}L_j(\theta_t)\right)\right\|^2\right] = \mathbb{E}\left[(4\alpha^2 G^2\eta_t^2)\left\|\frac{1}{M}\sum_{j=1}^{M}\widetilde{\nabla}L_j(\theta_t)\right\|^2\right]$$

$$\underset{(b)}{\leq}(8\alpha^2 G^2\eta_t^2)\mathbb{E}\left[\left\|\frac{1}{M}\sum_{j=1}^{M}\nabla L_j(\theta_t)\right\|^2 + \left\|\frac{1}{M}\sum_{j=1}^{M}\nabla L_j(\theta_t) - \frac{1}{M}\sum_{j=1}^{M}\widetilde{\nabla}L_j(\theta_t)\right\|^2\right]$$

$$\underset{(c)}{\leq}(8\alpha^2 G^2\eta_t^2)\left\{\mathbb{E}\left[\left\|\frac{1}{M}\sum_{j=1}^{M}\nabla L_i(\theta_t)\right\|^2\right] + B^2\right\}$$

where $(a)$ comes from Assumption 3, $(b)$ is due to $\|a + b\|^2 \leq 2\|a\|^2 + 2\|b\|^2$, and $(c)$ is by Assumption 2.

By applying the aforementioned bounds of $\mathbb{E}\left[\langle\nabla L(\theta_t), \theta_{t+1} - \theta_t\rangle\right]$ and $\mathbb{E}\left[\|\theta_{t+1} - \theta_t\|^2\right]$ to (1), we obtain:

$$\mathbb{E}\left[L(\theta_{t+1}) - L(\theta_t)\right] \leq \mathbb{E}\left[\langle\nabla L(\theta_t), \theta_{t+1} - \theta_t\rangle\right] + \frac{\beta}{2}\mathbb{E}\left[\|\theta_{t+1} - \theta_t\|^2\right]$$

$$\leq \mathbb{E}\left[(-\alpha\eta_t G^2)\left\{\|\nabla L(\theta_t)\|^2 + \|\frac{1}{M}\sum_{j=1}^{M}\nabla L_j(\theta_t)\|^2\right\}\right.$$

$$\left. + (4\alpha^2\beta G^2\eta_t^2)\left\{\left\|\frac{1}{M}\sum_{j=1}^{M}\nabla L_j(\theta_t)\right\|^2 + B^2\right\}\right]$$

$$= -\alpha\eta_t G^2\mathbb{E}\left[\|\nabla L(\theta_t)\|^2\right]$$

$$\underbrace{+ \alpha\eta_t G^2(4\alpha\beta\eta_t - 1)\mathbb{E}\left[\left\|\frac{1}{M}\sum_{j=1}^{M}\nabla L_j(\theta_t)\right\|^2\right]}_{\leq 0 \text{ if we choose } \eta_t \leq \frac{1}{4\alpha\beta}} + (4\alpha^2\beta G^2\eta_t^2)B^2$$

$$\leq -\alpha\eta_t G^2\mathbb{E}\left[\|\nabla L(\theta_t)\|^2\right] + (4\alpha^2\beta G^2\eta_t^2)B^2$$

Eventually, through the telescoping sum for $t = 0, 1, ..., T-1$, we obtain

$$L^* - \mathbb{E}\left[L(\theta_0)\right] \leq \sum_{t=0}^{T-1}(-\alpha\eta_t G^2)\mathbb{E}\left[\|\nabla L(\theta_t)\|^2\right] + \sum_{t=0}^{T-1}(4\alpha^2\beta G^2\eta_t^2)B^2$$

where $L^*$ represents the minimum value of $L(\theta)$.

After performing division on both sides by $H_T = \sum_{t=0}^{T-1}\eta_t$, and employing some manipulations, we obtain

$$\frac{1}{H_T}\sum_{t=0}^{T-1}\eta_t\mathbb{E}\left[\|\nabla L(\theta_t)\|^2\right] \leq \frac{\mathbb{E}\left[L(\theta_0)\right] - L^*}{\alpha G^2 H_T} + 4\alpha\beta B^2\left(\frac{1}{H_T}\sum_{t=0}^{T-1}\eta_t^2\right) \qquad (2)$$

By utilizing a decreasing learning rate (e.g., $\eta_t = \frac{\eta_0}{1+t}$), it can be seen that $H_T = \sum_{t=0}^{T-1}\eta_t \to \infty$ as $T$ increases, while $\sum_{t=0}^{T-1}\eta_t^2 < \infty$. Consequently, the upper bound stated in Equation (2) approaches 0 as $T$ grows, ensuring convergence towards a stationary point.

# D  Additional Experimental Result

In this section, we delve into the impacts of various parameters on both the training and communication rate. We first study the role of population size and the number of clients. Subsequently, we investigate the effect of additional compression techniques, such as sparsification, ranking, and quantization, on the model's performance. Lastly, we assess the efficacy of partitioning on clients in attaining better accuracy, and its relationship with the population size.

## D.1  Fitness Vector Sparsification (Top-k Subset Selection)

In this section, we explore the effect of fitness sparsification i.e. selecting top-k fitness values from the fitness vector of the whole population based on magnitude. We examined the effects of sparsification on two distinct population sizes: 128 and 1024. Without any sparsification, both populations demonstrated comparable performance. However, when we select the top-k most fit values, the denser population (comprising 1024 members) could tolerate a higher degree of sparsification compared to the less populous one (with 128 members).

To enable a fair and insightful comparison between the two population sizes, our focus was on assessing performance based on the number of members remaining post-sparsification rather than directly contrasting sparsification rates. We placed particular emphasis on the best and worst performing members, as they exert the most significant influence on the model update process in ES.

Fig. 1(a) and (b) visualize the sparsification process for populations of 128 and 1024, respectively, illustrating the performance decline that occurs as the number of remaining members diminishes.

Fig. 1(c) provides further insights into the performance improvements achieved by selecting top-8 or top-16 members from the initial set of 128 or 1024, as compared to optimizing with the whole population of 8 or 16.

Our results underline the crucial role that population size plays in exploring optimal solutions, overshadowing even the significance of compression rate. A larger population allows for broad exploration that can later be compressed to a smaller number of members without a performance loss. However, initiating the process with a smaller population cannot achieve equivalent performance due to the restricted exploration. Therefore, population size is a critical factor affecting the efficacy of exploration in evolutionary strategies.

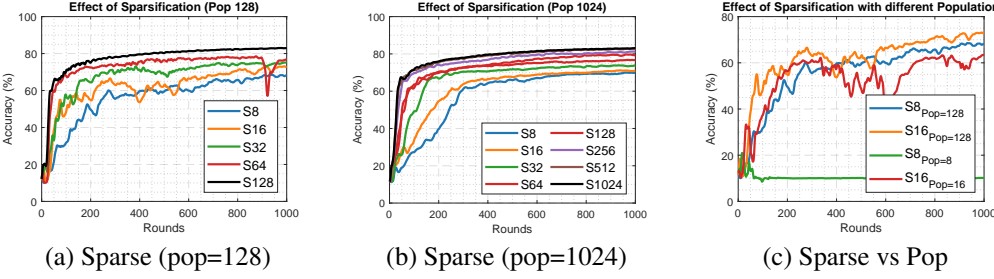

(a) Sparse (pop=128)  (b) Sparse (pop=1024)  (c) Sparse vs Pop

Figure 1: Effect of sparsification on EvoFed

## D.2 Ranking and Quantization

In this section, we examine the sensitivity of EvoFed to the precise value of fitness. We propose two techniques to reduce the bits required to represent the fitness vector, thus enhancing compression without compromising performance. For a clearer understanding of these methods' impacts, we chose a population size of 32, which is relatively less populated and has minimal redundancy, highlighting the insensitivity of EvoFed to precise fitness values.

Fig. 2(a) depicts the effect of quantization with varying bit numbers. The legend represents the number of bits used for quantization as a numeral followed by the letter $Q$, where $Q32$ indicates no compression and $Q1$ signifies transmitting a single bit (either 0 or 1) in place of the fitness value. The result exhibits a marginal performance loss even with $Q2$, illustrating EvoFed's insensitivity to precise fitness values and the potential for further compression gains through quantization.

Fig. 2(b) presents the performance when we transmit the member's rank within the population instead of the fitness value. In the legend, the number of samples assigned the same rank is denoted as a numeral following the letter $R$; $R32$ indicates assigning 32 different ranks to all members, and $R1$ implies assigning the same rank to every member. This ranking technique, a common practice in the Evolutionary Strategies literature, is typically employed when fitness values derived from the environment are noisy, and the quality of the solution can be improved by transmitting the ranking instead. However, where we have high-quality fitness measures derived from L2 loss, this technique only slightly improves the performance while reducing compression gains. By assigning the same rank to neighbouring samples within the fitness ranking, we can further enhance compression performance.

Comparing ranking and quantization, it is observed that quantization delivers superior performance with the same number of bits. Additionally, the number of bits used in quantization is independent of the population size, making quantization a more appropriate approach for compressing fitness values.

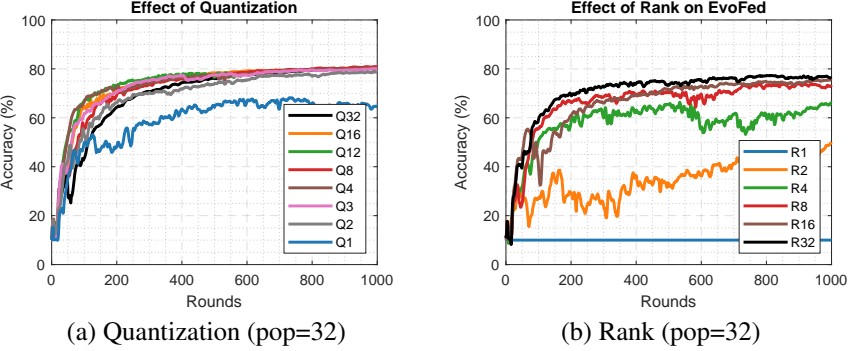

|           (a) Quantization (pop=32)           |           (b) Rank (pop=32)           |

Figure 2: Effect of Quantization on EvoFed

## D.3 Partitioning

The EvoFed's partitioning technique, as described in the main text, features a unique attribute that enhances performance. This technique maintains a fixed number of population samples at each client, thereby addressing memory limitations on the clients but necessitating increased communication as a trade-off. Although sparsification results underscore the importance of population size for exploration, partitioning presents an additional approach that navigates the limitation posed by the compression rate to improve performance.

Fig. 3 illustrates the impact of partitioning in four scenarios, each with a different population size. The results emphasize that partitioning is most effective when the clients cannot manage a sufficient number of samples to attain satisfactory performance. Partitioning enables us to gather more information from the limited sample size.

Each sub-figure in Fig. 3 includes baselines without partitioning, allowing for the comparison of improvements achievable either through an increased population size or the use of more partitions while maintaining a consistent communication rate. The legend of each figure specifies the number of partitions (the number following the letter $k$) and the population of the baselines (the number following the letter $p$). The volume of information required to be communicated for one round is also depicted for each method in the legend.

Fig. 3 clearly shows that using population sizes of 32 and 128 results in only a marginal improvement in performance. However, when utilizing population sizes of 8 and 16, a significant and noticeable improvement can be observed.

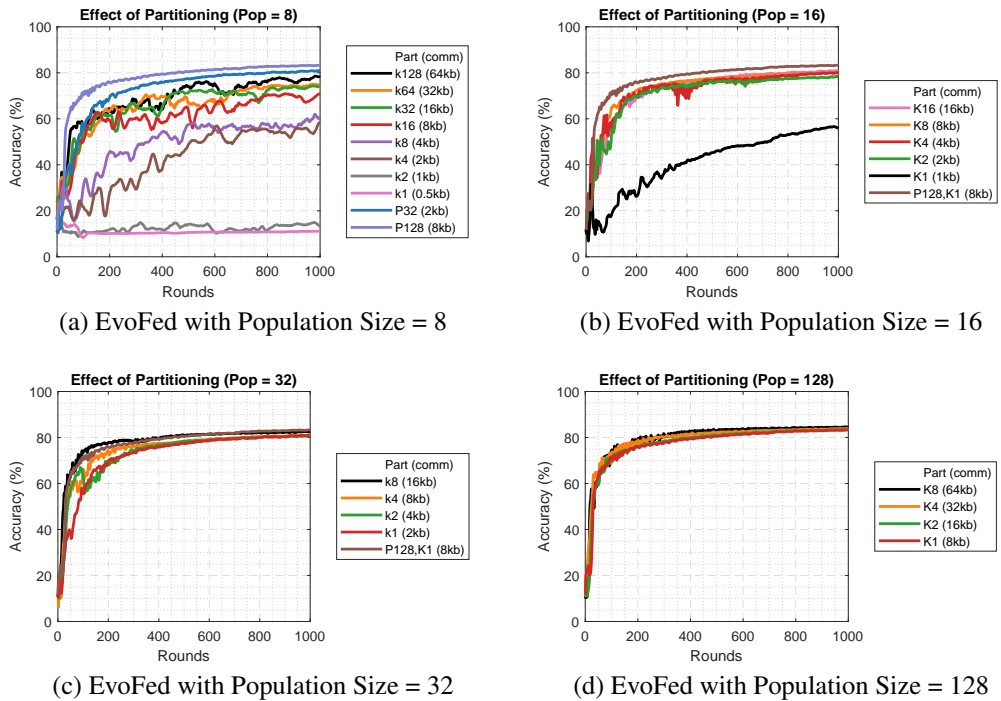

(a) EvoFed with Population Size = 8

(b) EvoFed with Population Size = 16

(c) EvoFed with Population Size = 32

(d) EvoFed with Population Size = 128

Figure 3: Effect of partitioning on EvoFed

### D.4 Larger Dataset and Model

Additionally, we investigate the efficacy of EvoFed in the context of a larger model and a bigger dataset. Fig. 4(a) showcases the performance on CIFAR-100 dataset with the same model parameters as those used in the CIFAR-10 experiment. The results show that EvoFed, although having a slower convergence rate, achieves higher performance than FedAvg eventually, with a significant compression rate. Fig. 4(b) illustrates the performance gain on CIFAR-10 dataset when the CNN layers are doubled. As the experimental result shows, having a larger model generally leads to better performance with slower convergence. EvoFed follows the same trend as BP in a centralized setting, suggesting the compression has not been affected by model size. All experiments were conducted with a population size of 32 and 50 partitions.

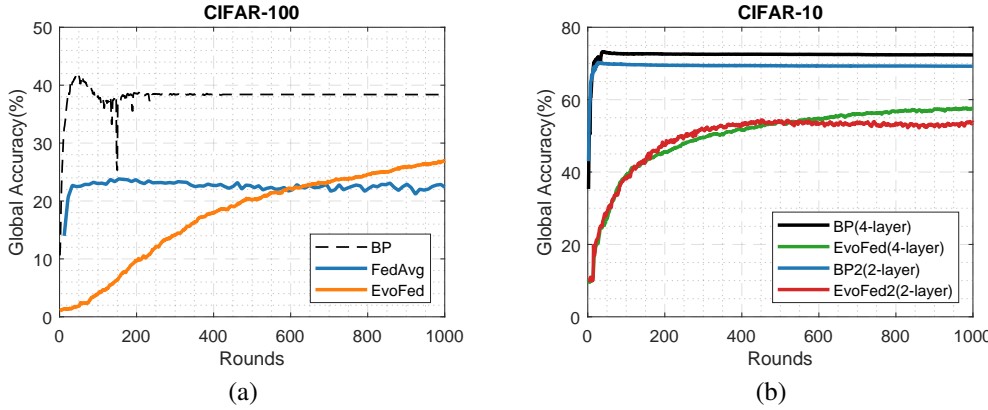

(a)

(b)

Figure 4: Larger dataset and Model: (a) shows performance on CIFAR-100, and (b) depicts the impact of having a larger model on CIFAR-10.