# OpenReview forum: "EvoFed: Leveraging Evolutionary Strategies for Communication-Efficient Federated Learning"
_NeurIPS.cc/2023/Conference — NeurIPS 2023 poster_

### Official Review · Reviewer_91Ta · 2023-06-27

**Soundness:** 3 good
**Presentation:** 3 good
**Contribution:** 3 good
**Rating:** 6
**Confidence:** 4

**Summary:**

This paper solves the huge communication burden problem in federated learning (FL). This paper proposes an evolutionary strategy (ES) based way to address the communication burden. Specifically, the ES only needs to transmit the fitness values among clients and server, and the fitness values could reflect the model update direction. The proposed EvoFed could compress the model more than 98.8% on FMNIST and 99.7% on CIFAR-10, nearly reducing all communication.

**Strengths:**

1. The proposed EvoFed is novel and interesting. Instead of transmitting models or model updates, the authors sample several model points with noise perturbation and then calculate fitness values which could reflect the gradient directions. Transmitting fitness values could save lots of communication cost.
2. The paper is sounded and theoretically supported. EvoFed is motivated by the population-based gradient estimation, and Equation 4 could derive the plausibility of utilizing the distance function as the fitness value function. Additionally, this paper also provide the convergence analysis to show the learning speed of population-based ES. According to Theorem 1, if the population size M is large enough, the second term of RHS could be sufficiently small.
3. Privacy enhancement via TEEs are considered.
4. The supp. provides abundant experimental studies including the population sizes and the partition ways.

**Weaknesses:**

1. The paper only studies small datasets such as FMNIST and CIFAR-10. These two benchmarks are easy to obtain good performances. More complex datasets such as CIFAR-100 or benchmarks in FedML should be investigated. On complex datasets, the reviewer thinks that a population size of 32 or 128 could not lead to qualified accuracy compared with FedAvg.
2. The reviewer sees that the local update epoch is one in section 4.2. However, updating local models up to more epochs should be studied and reported. If the local epoch becomes larger, the reviewer thinks that the population size should be larger.

**Questions:**

1. Some important experimental studies should be moved to the main body, e.g., the ablation study of population size, which is a significant factor in EvoFed.

**Limitations:**

The authors provide limitations and social impacts.

---

> ### Author Rebuttal · Authors · 2023-08-09
>
> > The paper only studies small datasets such as FMNIST and CIFAR-10. These two benchmarks are easy to obtain good performances. More complex datasets, such as CIFAR-100 or benchmarks in FedML, should be investigated.
>
> As other reviewers have also suggested, we have extended our experiments to include larger datasets. Specifically, we've evaluated our method on CIFAR100 using the same model we employed for CIFAR10. Our scheme exceeds FedAvg with sufficient rounds, as seen in Fig. 1 of the attached PDF.
>
>
> > On complex datasets, the reviewer thinks that a population size of 32 or 128 could not lead to qualified accuracy compared with FedAvg.
>
> Yes and no. Please again see Fig. 1 of the attached PDF, where we used the same population size of 32, the same as we used for CIFAR-10. For early rounds, EvoFed falls below FedAvg. But as the rounds progress, EvoFed eventually performs better.
>
>
> > The reviewer sees that the local update epoch is one in section 4.2. However, updating local models up to more epochs should be studied and reported. If the local epoch grows, the reviewer thinks the population size should be larger.
>
> We appreciate your attention to detail. When we described our algorithm, we simply talked about 1 local update for each global round, but we actually ran experiments with multiple local updates. For example, in Fig. 8 of the main paper. We will clarify the specification in the revision. We find that the population size does not have to grow as the local epoch increases, as the gradient estimation depends on the distance between the perturbation samples and the last updated model, no matter how many local updates are done.
>
>
> > Some important experimental studies should be moved to the main body, e.g., the ablation study of population size, which is a significant factor in EvoFed.
>
> We agree. We will move the key results mentioned to the main body during revision.

---

### Official Review · Reviewer_rYCA · 2023-06-28

**Soundness:** 2 fair
**Presentation:** 2 fair
**Contribution:** 2 fair
**Rating:** 6
**Confidence:** 4

**Summary:**

This paper proposes to utilize a variant of evolution strategy in federated learning framework in order to reduce its communication cost while maintaining the accuracy. The proposed approach, EvoFed, is mainly based on so called OpenAI-ES, a variant of evolution strategy intended to be used for policy optimization on decentralized environment. Differently from OpenAI-ES, which communicates the fitness of each candidate solution (cumulative reward in RL setting), EvoFed lets clients perform stochastic gradient descent locally with their local dataset and the fitness of candidate solutions are assigned based on the distance to the locally-updated parameter value. In this way, the gradient information is utilized and it is expected to improve the efficiency of the training, which is one of the most important disadvantages of evolution strategies. The authors performed a theoretical analysis of the proposed algorithm as well as empirical comparison with some baseline approaches on MNIST, FMNIST, and CIFAR10.


**Strengths:**

* a novel and promising application of evolution strategies to federated learning framework to reduce the communication cost.
* a novel fitness evaluation scheme for evolution strategies so that they can utilize stochastic gradient
* empirical results showing performance comparative to FedAvg, a standard federated learning approach, with reduced communication cost.


**Weaknesses:**

* limited empirical evaluation: both the image size and the network size are relatively small.
* theoretical bounds provided in the paper doesn’t probably reflect the reality.
* limited technical novelty


**Questions:**

1. (related to the 1st weakness pointed out above) the experiments are done on small image classification tasks. Because it was not clearly stated in the introduction what kind of tasks are considered as target tasks of this work, I couldn’t confirm whether these experiments reflects the target of FL. Moreover, the network size is relatively small. For MNIST and FMNIST, the number of parameters of a used CNN is only 11k. For CIFAR-10, it is increased to 2.3M, but the accuracy of the model trained by the standard back propagation was only about 70 %. It is interesting to see whether the proposed approach is still effective for greater models, where the communication cost is more important.

2. (related to the 2nd and 3rd weaknesses pointed out above) Related to Theorem 1, there are several questions.
2.1. It seems the statement in Theorem 1 contains some mistakes. In delta_M, i appears but delta_M is used as if it is independent of i. Is the summation or something is missing? Moreover, theta in delta_M is not defined. Is it theta_t? If so, is the summation or something is missing? Because of these points, it was not clear what is the point of the theorem.
2.2. Because of the above point, I am not sure if I understand it correctly. But it looks like that it doesn’t really implies the convergence as delta_M is irrelevant to T. Having a large M implies more communication cost. The intension of this Theorem should be more clearly explained in the main text.
2.3. The standard deviation is supposed to be proportional to the square root of T to derive the bound. If T is large, it means that the candidate solutions are far from the current center of distribution. In this situation, I coudn’t understand why it makes sense to define the fitness of the candidate solutions by the distance from the updated solution. This point should also be explained.
2.4. What is eta? The learning rate for SGD to get theta’? Please explain it.
2.5. Immediately after Theorem 1, the authors say “the appropriate value of alpha is 0.5 by the aforementioned reasoning.” However, I couldn’t get which part is refereed to.

3. The fitness is sometimes defined by the Euclidean norm, and sometimes its square. From the derivation I think it should be squared. I found this inconsistency in Eq (3) and Eq (6) and Algorithm 1. Please check this point carefully.

4. In Eq (7) b_j is introduced, but it is not used later in the paper. It seems that they are implicitly assumed to be equal. Please state it.

5. Figure 8.d, e, f are visible only for high accuracy part. I suggest to show these graph in log-scale so that the differences are visible also for the low accuracy part.


**Limitations:**

No limitation is explained in the paper.

---

> ### Author Rebuttal · Authors · 2023-08-09
>
> > limited empirical evaluation: both the image and network sizes are relatively small.
>
> Thanks for the comments; we have augmented our experiments to encompass larger datasets and networks. To disentangle the influences of task/dataset complexity from model architecture complexity, we undertook two specific experiments: We employed the same model used to run CIFAR10 on CIFAR100 and also fortified the model with two additional CNN layers. The comparative results of these investigations are depicted in Fig. 1 and 2 against backpropagation (BP, a centralized setup) and FedAvg.
>
> Fig. 1 reveals a generally subdued performance for both BP and FedAvg on the larger CIFAR100 dataset, yet EvoFed manages to align its performance closely with that of FedAvg and even exceeds eventually. In Fig. 2, the performance gap (with many rounds) between the 2 layers and 4 layers is the same between EvoFed and BP, confirming that EvoFed accuracy improves with the network size. What we want to stress here also is that we maintained the same population size as we went from 2 layers to 4 layers with EvoFed, meaning that the compression ratio actually improved with the 4-layer EvoFed.
>
>
> > Related to Theorem 1, there are several questions
>
> > 2.1. In delta_M, i appears but delta_M is used as if it is independent of i. Is the summation or something is missing? Moreover, theta in delta_M is not defined. Is it theta_t? If so, is the summation or something is missing? Because of these points, it was not clear what is the point of the theorem.
>
> > 2.2. Because of the above point, I am not sure if I understand it correctly. But it looks like that it doesn’t really implies the convergence as delta_M is irrelevant to T. Having a large M implies more communication cost.
>
> $\delta_M$ depends on $i$, yes. But $\delta_M$ can be suppressed to a small value with sufficiently large $M$, regardless of $i$. And yes, $\theta$ depends on $t$ as well, but again the sample mean estimator is close enough to the expectation across $t$ given large $M$, so that $\delta_M$ stays marginal regardless of $i$ or $t$. On the communication cost issue, we agree with the reviewer in principle, but empirically we find that EvoFed converges with fairly reasonable values of $M$ (e.g., $M=128$ ).
>
> > 2.3. The standard deviation is supposed to be proportional to the square root of T to derive the bound. If T is large, it means that the candidate solutions are far from the current center of distribution. In this situation, I coudn’t understand why it makes sense to define the fitness of the candidate solutions by the distance from the updated solution. This point should also be explained.
>
> We appreciate the comment. We first give a short answer: This condition on sigma was also unsettling to us, but we were able to empirically find practical $T$ values that were large enough to make the bound small while we could still find reasonable perturbation samples using the resulting sigma, albeit large. Another answer we would like to give is that we have managed to obtain an alternative bound that does not rely on the growing sigma condition. We were able to include this alternative bound in Appendix (see B.3 of Appendix, page 5). Given a chance, we hope to replace the main manuscript’s bound with this one.
>
> > 2.4. What is eta? The learning rate for SGD to get theta’?
>
> It is indeed the learning rate for SGD to obtain theta.
>
>
> > 2.5. Immediately after Theorem 1, the authors say “the appropriate value of alpha is 0.5 by the aforementioned reasoning.” However, I couldn’t get which part is refereed to.
>
> Eq.4 discusses how the gradient on the loss surface translates to the gradient on the convex fitness surface, utilizing $0.5$ as the learning rate. First, we refer to Eq.4. It is to emphasize that if we use a learning rate of $0.5$, the result of updating the model using the evolutionary strategy can be the same as the result of updating the model using backpropagation (i.e., $\theta - 0.5 \times \nabla_{\theta} |\theta - \theta'|^2 = \theta - 0.5 \times (2\theta - 2\theta') = \theta'
> $). We will make this clear in our revision.
>
>
> > The fitness is sometimes defined by the Euclidean norm, and sometimes its square. From the derivation I think it should be squared. I found this inconsistency in Eq (3) and Eq (6), and Algorithm 1. Please check this point carefully.
>
> Thank you; the correct fitness is the l2-square. We will ensure consistency throughout the document.
>
>
> > In Eq (7), b_j is introduced, but it is not used later in the paper. It seems that they are implicitly assumed to be equal. Please state it.
>
> Yes, the batch size is assumed to remain the same across clients. We will make this clear.
>
>
> > Figure 8.d, e, f are visible only for high accuracy part. I suggest to show these graph in log-scale so that the differences are visible also for the low accuracy part.
>
> Thank you for the suggestion; we will revise it accordingly. Our additional experimental results are shown in the log scale in the attached pdf.

---

> > ### Comment · Reviewer_rYCA · 2023-08-13
> > **Thank you for your response**
> >
> > I am satisfied with the authors response. The bound derived in the appendix makes more sense to me, and I agree that replacing the current theorem with the one in the appendix is beneficial.

---

> > > ### Author Response · Authors · 2023-08-14
> > > **Reply to Reviewer rYCA**
> > >
> > > Thank you for your prompt response; we will definitely do so.

---

### Official Review · Reviewer_cJ4W · 2023-07-06

**Soundness:** 2 fair
**Presentation:** 2 fair
**Contribution:** 2 fair
**Rating:** 6
**Confidence:** 3

**Summary:**

This paper offers a solution to a problem in federated learning: it's expensive to pass large model parameters back and forth from the server to the local/client nodes who train with their own data. The solution depends on the same random number generator and seed in every local node plus complete synchronization of model parameter updates. Rather than pass the model parms, distance measures (aka fitness scores) between  a population of perturbed models and a locally trained model are transmitted to the server, then aggregated and passed back to the clients who update the model parms locally.  This local (or server) model parameter update is the crux of the algorithm. The sum of all clients' distances for one perturbation model is tweaked according to an ES covariance matrix of perturbations  and added to the existing parameter with normalization and two  factors which are the learning rate parameter and inverse of the noise vector. This is repeated for every perturbation model in the local population.  This is called fitness-based information sharing and the method is called "Population-Based Gradient Estimation'.  The population of perturbations within an Evolutionary Strategy must be carefully chosen in terms of variance (sigma) and size (M).  Because every local node references the same perturbed models and  the distances reflect the local data at each node, there is a claim of nearly full information encoding while the transmission cost of the distances is much smaller that then number of model parms.  The depth of the model update claim comes from showing that the summed differences of each client perturbation model  *and*, summed over M perturbation models,  approximates a gradient parallel to the gradient trained by back propagation.  The ES handles the accurate addition of noise relative to the perturbations and their variance. A privacy enhancement through Fully Homomorphic Encryption and the clients being TEEs/Trusted Execution Environments is also provided.



**Strengths:**

This is a form of adaptive compression which is a great way to solve a communications overload problem. Most approaches transmit the updates to the model or gradient signals.  EvoFEd outperforms traditional compression methods. The L2 norm distance calculation is simple and so is the server-side batchsize-weighted aggregation and update. The client-side update of the model references noise vectors from the populations.  The integration with fully homomorphic encryption is good.

**Weaknesses:**

It's not entirely clear from the paper in what types of fitness landscape this will work. And, it's not clear why a parallel gradient which is then perturbed is going to lead to *optimal* convergence.
 A Natural Evolutionary Strategy is used to generate perturbations of the model's parms and the newly perturbed models are then used as benchmarks to measure a distance that reveals the gradient of one model that is trained with back propagation. The stochastic gradient ascent of NES is discarded. More information on the factored Gaussian parameter distribution that motivates its selection would be helpful.  An explanation of why the NES was chosen over other ES would improve clarity.
It's also not clear how to set any of the parameters in the NES, ie population (M) and variance.

This method requires more synchronization than a federated learning model in general.

**Questions:**

Is mirrored sampling important?
What defines convergence?
What role does batch size play in the algorithm's efficiency? Are clients expected to draw different batches after each update from the server?
How are parameters chosen?

**Limitations:**

no limitation in this scope

---

> ### Author Rebuttal · Authors · 2023-08-09
>
> > It's not entirely clear from the paper in what types of fitness landscape this will work. Why would a parallel gradient, when perturbed, lead to optimal convergence?
>
> We appreciate the reviewer raising an important question. In a nutshell, as long as the score function estimator of Eq.2 is correct (which would be the case with a sufficiently large population; see Appendix C.1), the estimated gradient $\frac{1}{\sigma} \nabla_{\theta} \mathbb{E}_{\epsilon \sim \mathcal{N}(0,1)} { f(\theta + \sigma \epsilon) \epsilon }$ is parallel to the true gradient of the loss function $\nabla\_\theta L(\theta_t)$.
> More specifically, if we choose L2 distance as the fitness function as done in the paper, the evolution-strategy-based model update becomes identical (assuming a learning rate of 0.5) to the backpropagation-based model update (as for the proof, please refer to B.1 in Appendix).
>
> > The stochastic gradient ascent of NES is discarded.
>
> Actually, the update step that happens on the aggregation of fitness vectors is the gradient ascent of NES. We tuned a separate optimizer for this and provided the hyperparameters in Appendix D.
>
> >More information on the factored Gaussian parameter distribution that motivates its selection would be helpful.
>
> Factored Gaussian has a diagonal covariance matrix, leading to reduced memory usage. This benefit is especially important in our FL application, where memory resources are critical to all individual clients.
>
>
> > An explanation of why the NES was chosen over other ES would improve clarity.
>
> Natural Evolution Strategies (NES) are typically chosen when there's a large difference between the number of parameters and the population size, especially when our objective isn't to obtain a wide range of solutions. NES is effective at estimating a single gradient because it uses uniform perturbations. The straightforward nature of OpenAI-ES not only makes it efficient (in terms of memory and CPU demands) and simplifies our convergence analysis. Plus, there's no need to manage extra covariance matrices. When comparing OpenAI-ES with other methods like CMA, PGPE, and PSO, we found that OpenAI-ES had the upper hand, as illustrated in Fig.3 of the attached PDF.
>
> > It's also not clear how to set any of the parameters in the NES, ie population (M) and variance.
>
> We suggest adjusting parameters by gradually increasing the population size until reaching an optimal point, as described in Appendix C.1. Further parameter adjustments can be achieved through Bayesian search.
>
>
> > This method requires more synchronization than a federated learning model in general.
>
> When we described our algorithm in the main manuscript, we simply spoke of one local iteration per global round, but in actual experiments, multiple local updates are made for each global exchange (this will be clarified in the revision). In this sense, the synchronization requirement is no different from other FL schemes. Now, even if the reviewer means seed synchronization, the seeds can be predetermined, and again the requirement is the same as in conventional FL, where the initial models must be shared among all participants.
>
> > Is mirrored sampling important?
>
> Mirror sampling is indeed very important. Mirrored sampling ensures an unbiased population. As demonstrated in [2], mirrored sampling eliminates the impact of outlier individuals and prevents ES from falling into local optima during the initial stages of training. It plays a pivotal role in theoretical analyses, especially concerning sigma, and sigma^3 sums up to zero.
>
> [2] Salimans, Tim, et al. "Evolution strategies as a scalable alternative to reinforcement learning." arXiv preprint arXiv:1703.03864 (2017).
>
>
> > What defines convergence?
>
> Convergence in the context of neural networks is achieved when the training process stabilizes and the network's parameters no longer significantly improve the model's performance. In the case of our theorem, we proved that the gradient of the loss function with respect to the parameters $(\nabla L(\theta))$ converges to $0$ as the communication round $t$ increases. Please see section B.3 of the Supplementary Material for more detail.
>
>
> > What role does batch size play in the algorithm's efficiency?
>
> First, it's important to note that gradient estimation using ES remains consistent regardless of the batch size used to obtain the target gradient. The roles of batch sizes mirror traditional learning but do not influence compression rates in our setting because compression is done after getting the target update.
>
> > Are clients expected to draw different batches after each update from the server?
>
> In our experiments, we select different batches after each update aligned with standard training practices.
>
> > How are parameters chosen?
>
> Optimal parameters are determined through a Bayesian search by running more than 200 experiments for the initial 500 rounds.

---

> > ### Comment · Reviewer_cJ4W · 2023-08-17
> >
> > Thank you for your clarifications. I am satisfied with your responses.

---

### Official Review · Reviewer_3tMK · 2023-07-09

**Soundness:** 3 good
**Presentation:** 2 fair
**Contribution:** 3 good
**Rating:** 6
**Confidence:** 3

**Summary:**

This paper introduces a new method called EvoFed, which combines Evolutionary Strategies (ES) with Federated Learning to address the challenge of high communication costs in the traditional FL setting. Instead of sending the actual model parameters, each device sends a measure of how similar its updated model is to a set of perturbed models. These perturbed models are generated in the same way on every device and server. By carefully choosing the variance and size of the perturbed models, the similarity measures can carry almost all the information about the model parameters. Since the perturbed models are smaller in size, the communication load is reduced. The authors show results in a deep CNN network trained on MNIST, FMNIST and CIFAR-10, demonstrating that EvoFed performs as well as FedAvg and compresses the model by at least 98%.

**Strengths:**

1. The idea of using evolutionary schemes for model-sharing in FL is compelling and seems original.
2. The work could have high impact (if evaluated thoroughly) and pave way for large-scale adoption of FL on edge applications.


**Weaknesses:**

1. The paper is difficult to follow at places. For instance, authors need to clarify their intuition for model partitioning in Section 3. How are the number of partitions chosen? Is this model-agnostic? If not, some empirical results or insights can help the reader reproduce the findings in their works.
2. The paper has some hard to follow sentences and uses abbreviations that are not expanded prior (e.g., BP). A thorough proof-reading of the manuscript would be helpful.
3. The evaluations are limited. The paper introduces a new paradigm for federated learning using evolutionary schemes which reduces communication costs significantly. In order to understand the true impact of this idea, it needs to be juxtaposed against some state-of-the-art pruning techniques for FL (e.g., Lin, Rongmei, Yonghui Xiao, Tien-Ju Yang, Ding Zhao, Li Xiong, Giovanni Motta, and Françoise Beaufays. "Federated pruning: improving neural network efficiency with federated learning." arXiv preprint arXiv:2209.06359 (2022)).


**Questions:**

Please see comments above in weakness.

**Limitations:**

The paper could benefit from more discussion on the limitations of the proposed method. There is no obvious negative societal impact.

---

> ### Author Rebuttal · Authors · 2023-08-09
>
> > The authors need to clarify their intuition for model partitioning in Section 3.
>
> Evolutionary Strategies exhibit sensitivity to the ratio of the population to parameter size. As detailed in Appendix C.1, a population size that is not adequate can hinder convergence. To assist clients with resource constraints, we introduced model partitioning. By computing fitness over multiple partitions, gradient estimation accuracy (as presented in Eq. 2) can be improved even with a small population size. Therefore, by compressing manageable portions of the model with a small population, we can enhance convergence, as elaborated in Appendix C.4.
>
> >How is the number of partitions chosen?
>
> The number of partitions is tailored according to the available resources for clients. As delineated in Tables 1 and 2 of Appendix A.4, there's a linear correlation between increasing population and client resource usage. For resource-limited clients, elevating the number of partitions offers a viable strategy to uphold performance without overburdening resources. A detailed exploration of the effects of partitioning granularity at different population sizes can be found in Appendix C.4.
>
> > Is it model-agnostic? If not, empirical results or insights would help in reproducing the findings."
>
> Indeed, our partitioning approach is model-agnostic. We treat all models as data vectors, regardless of their built-in structure. This allows for versatile application across various model architectures.
>
> > The paper has unclear sentences and unexplained abbreviations (e.g., BP).
>
> We will take a thorough review to fix any ambiguities and make sure we're clear in our writing.
>
>
> > The paper should be compared with state-of-the-art pruning techniques for FL.
>
>
> We maintain that EvoFed is designed to integrate with other techniques, such as quantization or pruning, seamlessly. This implies that even state-of-the-art pruning methods can leverage the compression efficiency of EvoFed, further optimizing their performance. Having said that, we also provide a comparison with a SOTA technique. Unfortunately though, the paper that the reviewer suggested does not provide source codes, and the reported results there do not allow apple-apple comparison with ours either. Thus, we chose PruneFL [1], a similar paper, as our benchmark. We've detailed a comparison between PruneFL and EvoFed in Fig. 4 from the attached PDF. The results clearly show EvoFed’s reduction in communication needs compared to both PruneFL and FedAvg.
>
> [1] Jiang, Yuang, et al. "Model pruning enables efficient federated learning on edge devices." IEEE Transactions on Neural Networks and Learning Systems (2022).
>
> > There should be more discussion on the limitations of the proposed method.
>
>
> One constraint of our method pertains to the resource demands accompanying an enlarged population size. While we provide some practical ways to get around this issue (refer to Appendix A, where we analyze resource utilization in detail), the complexity of our method depends strongly on population size, and thus one has to be very cautious in designing the overall system. We will mention this in our revision.

---

> > ### Comment · Reviewer_3tMK · 2023-08-17
> >
> > Thank you for addressing my questions and the additional results with PruneFL. I am satisfied with the rebuttal and will raise my score.

---

> > > ### Author Response · Authors · 2023-08-20
> > >
> > > We are pleased to hear that you are satisfied with our response. Thank you for reconsidering your score.

---

### Author Rebuttal · Authors · 2023-08-09

We are deeply grateful to all the reviewers for their detailed analysis and insightful comments on our work. We appreciate the recognition of the novelty and potential impact of our approach, its promising application in reducing communication costs, the successful integration with encryption methods, and our substantial experimental studies.
We have carefully considered all the concerns raised by the reviewers and conducted additional experiments to address them. We provide answers to each reviewer below.

---

### Decision · Program_Chairs · 2023-09-21

**Decision:**

Accept (poster)

**Comment:**

This proposes EvoFed, a method that combines Evolutionary Strategies (ES) with Federated Learning to address the challenge of high communication costs in the traditional FL setting. Instead of sending the actual model parameters, each device sends a measure of how similar its updated model is to a set of perturbed models. These perturbed models are generated in the same way on every device and server. Also, EvoFed significantly reduces communication overhead and enhances privacy by providing protection against eavesdroppers who do not participate in collaborative learning.

All the reviewers (also myself) are in favor of accepting the paper. All the major concerns have been resolved after the rebuttal/discussion period.